# Integrating Hydrography Observations and Geodetic Data for Enhanced Dynamic Topography Estimation

Mahmoud Pirooznia [1], Behzad Voosoghi [1], Davod Poreh [2,3],* and Arash Amini [1]

1. Faculty of Geodesy and Geomatics Engineering, K. N. Toosi University of Technology, Tehran 15433-19967, Iran; ma.pirooznia@email.kntu.ac.ir (M.P.); vosoghi@kntu.ac.ir (B.V.); a_amini@email.kntu.ac.ir (A.A.)
2. Electrical Engineering Department, Sharif University of Technology, Tehran 14588-89694, Iran
3. Department of Electrical Engineering and Information Technology, University of Napoli Federico II, 80138 Naples, Italy
* Correspondence: davod.poreh@unina.it

**Abstract:** Dynamic topography (DT) refers to the time-varying component of the sea surface height influenced by factors like ocean currents, temperature, and salinity gradients. Accurate estimation of DT is crucial for comprehending oceanic circulation patterns and their impact on climate. This study introduces two approaches to estimating DT: (1) utilizing satellite altimetry to directly observe sea surface height and (2) considering the steric and non-steric components of sea level anomalies. The steric term is calculated using salinity and temperature data obtained from local buoy data, Argo observations, and the World Ocean Atlas model. The non-steric term is calculated using GRACE Satellite gravimetry data. To estimate the assimilated DT, four methods are utilized, including variance component estimation (VCE), Bayesian theory, Kalman filter, and 3D variational (3DVAR). These methods assimilate the two aforementioned schemes. The validity of the estimated DT is assessed by comparing the calculated sea surface current, derived from the obtained DT, with observations from local current meter stations. The results indicate that the VCE method outperforms other methods in determining the final DT. Furthermore, incorporating the steric and non-steric terms of sea level in determining DT in coastal areas enhances the accuracy of estimating sea surface currents.

**Keywords:** assimilation; dynamic topography; geodetic data; satellite altimetry; sea surface height; steric and non-steric

## 1. Introduction

Dynamic topography (DT) represents the variation of the sea surface height due to ocean circulation and currents, as well as other factors, such as tides and atmospheric pressure [1]. DT plays a significant role in oceanographic studies, affecting ocean circulation, heat transport, and the distribution of marine resources. Traditional approaches to computing DT rely on fixed reference surfaces, thus neglecting important temporal variations [2]. By incorporating in situ hydrography observations and geodetic data such as satellite altimetry through data assimilation approaches, a more accurate representation of DT may be achieved. Data assimilation is a technique used to combine multiple sources of data. Each type of observation provides valuable but limited information about the ocean's state and DT. By assimilating these diverse data sources, the strengths of each observation can be leveraged while compensating for their individual limitations. This can lead to a more complete and accurate estimation of DT [3]. By accurately estimating DT, we can gain valuable insights into these processes, further enhancing our comprehension of the intricate interplay between the ocean and the broader climate dynamics [4,5].

DT can be obtained via hydrographic data and the Gravity Recovery and Climate Experiment (GRACE). The computation of DT involves adding the sea level anomaly (SLA) to the mean dynamic topography (MDT), expressed as DT = SLA + MDT. The SLA

comprises two components—namely, the steric and non-steric components—which can be combined to derive the SLA [6]. The steric component of sea level can be determined using observed temperature and salinity data, as well as data from the global array of profiling floats (ARGO) and the World Ocean Atlas (WOA) model. Local sampling platforms may provide high-precision information, but there are limitations in accessing these areas, and currently, there is a lack of long-term and consistent data [7,8]. On the other hand, the non-steric component of sea level can be derived from observations obtained from GRACE twin satellites [9,10]. The MDT (MDT = MSS − N), representing the permanent stationary component of DT, is calculated as the difference between the local or global sea surface height (SSH) averaged over an extended period (referred to as the mean sea surface (MSS), which is available from global or regional models [11]) and the geoid undulation (N).

Altimetry observations can serve as an alternative dataset for computing DT and evaluating surface currents [12]. In this approach, DT can be obtained simply by subtracting the geoid undulation from the SSH measured by altimeter satellites with respect to the reference ellipsoid (DT = SSH − N). N can be extracted from marine gravity models derived from satellite-based gravity field models. Therefore, the precise determination of the geoid plays a crucial role in calculating DT in previous studies [13–15]. Although satellite altimetry missions estimate the sea surface height with high accuracy, there are still some issues that need to be considered in the preprocessing of SSH determination. One of the problems is radial errors, which are caused by the lack of precise modeling of environmental and geophysical corrections, as well as satellite orbit errors, requiring calibration [7]. Another issue in satellite altimetry missions is in coastal regions. The accuracy of satellite altimetry measurements in coastal regions relies on the waveform [16]. In oceans and open waters, the waveforms follow a model called Brown, but in coastal areas, enclosed waters, and rivers, due to factors such as shallow depth, surface fluctuations, ice cover, and unbalanced atmospheric conditions, this model changes because the transmitted wave interacts with a more complex surface, leading to errors in calculating the range, which is the distance among the satellite and the sea surface that requires waveform retracking corrections [17].

Thus, the estimation of DT is a complex task due to various factors, such as the limited spatial and temporal resolution of the data, errors and biases in the data, uncertainties in models, limited data coverage, and nonstationary signals [18].

In this paper, we focus on determining DT using satellite altimetry and hydrographic observations, as well as the assimilation of these two different schemes using data assimilation techniques to improve the estimation of DT in the study area and overcome the aforementioned problems. We aim to explore the potential of this assimilation technique for enhancing our understanding of the oceanographic dynamics in these regions, as well as its implications for various applications. By assimilating these datasets, we can improve our ability to monitor and predict changes in sea surface topography, contributing to better oceanographic modeling and management efforts in the study area.

The research conducted in this study specifically targets the Persian Gulf and the Oman Sea as its primary areas of focus. The Persian Gulf and the Oman Sea, located in the southwestern part of Asia, are two interconnected bodies of water that hold significant hydrographic features and play a vital role in the region's oceanographic dynamics. Understanding the dynamic characteristics and variations of the sea surface topography in these regions is of great importance for various applications, including navigation, climate studies, and coastal management [19].

The connection between DT and sea surface currents arises from the fact that the flow of water in the ocean, driven by currents, contributes to the generation of DT. As currents move water masses, they induce variations in SSH, resulting in spatial gradients in DT. This relationship is complex and plays a significant role in understanding the dynamics of the ocean system [12,19]. Certain studies have explored the estimation of surface currents using geodetic or hydrographic observations. Knudsen et al. (2011) computed a global MDT and ocean circulation using a preliminary gravity field and steady-

state ocean circulation explorer (GOCE) data. Their findings indicated that calculating global geostrophic surface currents from the MDT led to improvements in major current systems, such as the Gulf Stream, Labrador Current, and Greenland Current, particularly in the North Atlantic [14]. Bingham et al. (2014) utilized GOCE measurements to obtain steady-state surface circulation in the North Atlantic and compared it with drifter-based estimates. They discovered that, with proper filtering, GOCE could recover around 70% of the Gulf Stream strength compared with the best drifter-based estimates [20]. Additionally, Chang et al. (2016) estimated global surface and subsurface geostrophic currents using satellite altimetry and hydrographic data. The results demonstrated that combining satellite altimetry and hydrographic data for geostrophic current estimation yielded good agreement with in situ current meter observations, particularly in the meridional and North Atlantic regions [21]. Hence, in this study, akin to previous research [13,22], the utilization of in situ current meter data has been employed to identify the most effective data assimilation methods for estimating DT. Therefore, the current meter data serves as a fitting source for this purpose.

## 2. Materials and Methods

### 2.1. Data Description

The dataset consists of satellite altimetry measurements, the gravity field model used to determine geoid height, the GRACE satellite data, temperature and salinity observations obtained through direct measurements, and the World Ocean Atlas (WOA) model, as well as Ekman currents data. Table 1 presents the data source for satellite altimetry employed to calculate the sea surface height (SSH). The SSH computation relies on altimetry observations from multiple missions, specifically utilizing the sensor geophysical data record (SGDR), which contains valuable along-track waveform information. The altimeter waveforms may experience corruption near coastal regions, leading to biases in range measurements. To address this issue, a waveform retracking method known as the adaptive leading edge sub-waveform (ALES) is applied to correct the range observations [16]. Additionally, inherent uncertainties exist in satellite altimetry measurements, including errors introduced by geophysical and environmental factors, which are typically addressed through global or local modeling techniques [23,24]. Moreover, the systematic errors, such as radial error [24], present in the satellite altimeter datasets require calibration, which is considered as described in [25]. Given the limited spatial resolution of satellite altimetry in the cross-track direction, despite the utilization of multiple repeat-track or multi-mission data, there is a chance that certain regions may still be overlooked or not adequately captured in satellite altimetry measurements [26]. This limitation arises from the nadir measurement principle and the orbital characteristics of altimeter satellites, and it should not be disregarded [16]. To address this drawback, a least square collocation algorithm, as described in [25], is employed to interpolate SSH values at the missing points. The interpolation method employed in this study proves to be effective in generating missing sea surface height (SSH) values with an accuracy of 3–5 cm, which aligns with the typical nominal accuracy of satellite altimetry [19,27]. This indicates that the interpolated values closely match the expected precision of the satellite altimetry measurements.

**Table 1.** Description of altimeters used in this study.

| Missions | Cycles | Periods | Sources | Accuracy Value |
|---|---|---|---|---|
| Jason 1 | 001-259 | 15 January 2002–16 January 2009 | NASA, AVISO | about 4 cm |
| Jason 2 | 001-303 | 4 July 2008–1 October 2016 | AVISO | about 4 cm |
| Jason 3 | 001-050 | 18 February 2016–12 June 2017 | AVISO | about 4 cm |
| Envisat | 008-093 | 23 July 2002–18 October 2010 | ESA | about 3 cm |
| Saral | 001-035 | 14 March 2013–16 June 2016 | AVISO | about 8 cm |
| Sentinel3A | 001-083 | 16 March 2016–3 January 2023 | ESA | about 3 cm |
| Sentinel3B | 001-057 | 4 June 2018–13 January 2023 | ESA | about 3 cm |

In the following, six gravity models are examined for determining the geoid height and calculating DT. The specifications of these gravity models are provided in Table 2. The datasets utilized in model development are summarized in a data column, with the symbol "S" representing satellite data (such as GRACE, GOCE, and LAGEOS), "A" denoting altimetry data, and "G" indicating ground data (including terrestrial, shipborne, and airborne measurements). Additionally, the GRACE and GRACE Follow-On satellite data, respectively, for the time periods of 2002 to 2017 and 2018 to 2022 are used to determine the non-steric component of sea level; these data are obtained from the GFZ website. By utilizing the error propagation toolbox pioneered by [28], the errors related to the geoid heights in the study area are estimated at various spatial resolutions, following the Gaussian error propagation method [14,20]. The estimation is based on the full error covariance matrices of the gravity models. The XGM2019e geoid model exhibits the lowest error, particularly at higher resolutions. Consequently, this model is chosen to calculate the geoid height within the study area. The MDT and its derived DT are determined using the MSS-IR01 model, which is the local MSS utilized for this purpose. The MSS-IR01 model was developed by the National Cartography Center (NCC) of Iran, as outlined by [24].

**Table 2.** Gravity models description.

| Numbers | Models | Produced Year | Degree Count | Data Source | References |
|---|---|---|---|---|---|
| 1 | SGG-UGM-1 | 2018 | 2159 | EGM2008, S(GOCE) | [29] |
| 2 | EIGEN-6S4 (v2) | 2016 | 300 | S(GOCE), S(GRACE), S(LAGEOS) | [30] |
| 3 | GOCO05c | 2016 | 720 | A, G, S | [31] |
| 4 | GGM05C | 2015 | 360 | A, G, S(GOCE), S(GRACE) | [32] |
| 5 | EIGEN-6C4 | 2014 | 2190 | A, G, S(GOCE), S(GRACE), S(LAGEOS) | [33] |
| 6 | XGM2019e | 2019 | 5399 | A, G, S(GOCO06s), T (Topography) | [34] |

The steric component of sea level, which is influenced by salinity and temperature, is primarily determined using data from hydrographic measurements, the ARGO instrument, and the WOA model. Salinity and temperature data for coastal waters around the Bushehr Peninsula (12 stations from 28.7°N to 29°N and 50.65°E to 50.95°E) from July 2011 to July 2012, Chabahar Bay (25 stations from 25.2°N to 25.4°N and 60.4°E to 60.6°E) from January 2007 to March 2010, and Pozm Bay (17 stations from 25.2°N to 25.4°N and 60.4°E to 60.6°E) from October 2011 to August 2012 were obtained from the Iranian National Institute for Oceanography (INIO) hydrographic measurements.

Quality control procedures, in line with the Intergovernmental Oceanographic Commission (IOC) method, are applied to ensure data reliability. The ARGO data, available from the official ARGO website, consists of salinity and temperature profiles collected by a network of profiling buoys. The temperature and salinity measurements from Argo floats and buoys typically have an accuracy of around 0.002 degrees Celsius (°C) and, typically, around 0.01 practical salinity units (PSU), respectively. These buoys capture both seasonal and transient signals and cover a portion of the Sea of Oman from 2002 to 2016. In cases where salinity and temperature data are unavailable from hydrographic measurements or ARGO, the study makes use of the WOA model, which was developed by the Ocean Climate Laboratory of the National Oceanographic Data Center. The WOA model offers monthly gridded data points of temperature and salinity covering the years 2002 to 2022. These data points are computed using the international equation of state for seawater [35]. The WOA model can be accessed through the website of the National Oceanographic Data Center.

Furthermore, the assimilation methods used to determine dynamic topography (DT) are validated by incorporating local current meter observations from various stations. The specifications of these current meter stations are presented in Table 3. The data from these

stations are acquired from the Ports and Maritime Organization (PMO) of Iran. The ADCP has an accuracy of approximately 6 cm/s [36].

**Table 3.** Current meter station description.

| Number | Region | Equipment | Locations (Lat, Lon) | Periods | Sources |
|---|---|---|---|---|---|
| 1 | Khuran | ADCP | 26.7, 55.45 | 30 August 2005–10 April 2005 | INIO |
| 2 | Konarak | ADCP | 25.37, 60.43 | 21 August 2006–9 March 2007 | PMO |
| 3 | Chabahar | ADCP | 25.29, 60.47 | 21 August 2006–9 March 2007 | PMO |
| 4 | Bushehr | ADCP | 28.97, 50.66 | 15 June 2010–26 July 2011 | PMO |
| 5 | Taheri | ADCP | 27.63, 52.36 | 23 August 2008–24 September 2009 | PMO |
| 6 | Nayband Gulf | ADCP | 27.42, 52.65 | 5 November 2009–7 December 2009 | PMO |
| 7 | Nakhl Taghi | ADCP | 27.49, 52.57 | 22 August 2008–24 September 2009 | PMO |
| 8 | Kangan | ADCP | 27.83, 52.04 | 23 August 2008–25 September 2009 | PMO |
| 10 | Jask | ADCP | 25.65, 57.76 | 16 July 2010–23 January 2011 | PMO |
| 11 | Larak | ADCP | 26.82, 56.37 | 10 June 2009–10 December 2010 | PMO |
| 12 | Googsar | ADCP | 25.60, 57.77 | 7 December 2010–28 October 2010 | PMO |
| 13 | Rajaei | ADCP | 27.07, 56.08 | 10 December 2009–1 December 2010 | PMO |

Thorough quality control measures have been implemented on all the data utilized in this study, ensuring the removal of erroneous observations and conducting outlier analysis, among other techniques. For detailed information regarding the specific methods employed for quality control of the observations, please refer to the work of [37].

### 2.2. Determination of DT Using Two Different Schemes

As previously stated, satellite altimetry is a method employed to calculate *DT*. The corresponding equation, as presented by [19], is as follows.

$$DT(\lambda, \varphi, t) = SSH(\lambda, \varphi, t) - N(\lambda, \varphi) \tag{1}$$

Here, *SSH* is the sea surface height that refers to the elevation of the ocean's surface relative to a reference ellipsoid and *N* is the geoid height.

Another method of estimating *DT* involves utilizing the steric and non-steric components of sea level anomaly, as outlined in the subsequent equation [38]. Figure 1 illustrates the relationship between Equations (1) and (2) in determining *DT*.

$$DT(\lambda, \varphi, t) = SLA(\lambda, \varphi, t) + MDT(\lambda, \varphi)$$
$$MDT(\lambda, \varphi) = MSS(\lambda, \varphi) - N(\lambda, \varphi) \tag{2}$$

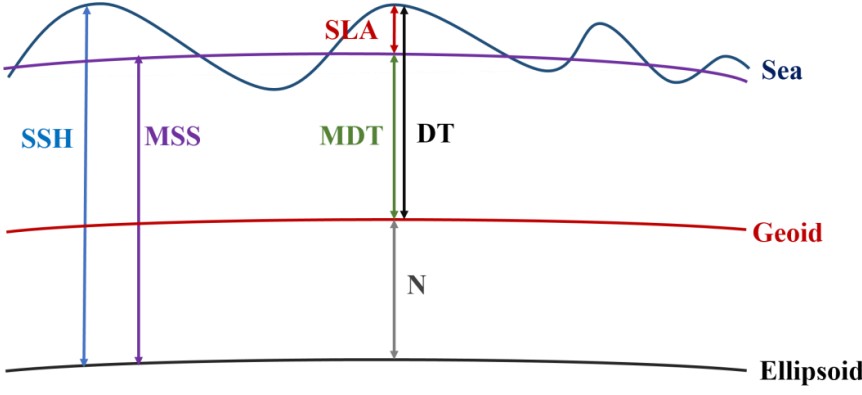

**Figure 1.** The relationship between oceanic parameters in determining DT.

In the provided equation, *SLA* denotes sea level anomaly, *MDT* signifies mean dynamic topography, and *MSS* corresponds to the mean sea surface. The determination of *SLA* involves the combination of steric and non-steric components, expressed mathematically as depicted [10]. The steric component of sea level refers to changes in seawater density caused by variations in salinity and temperature. On the other hand, the non-steric component is primarily associated with the movement of water masses across oceans, land, and the atmosphere [6,9].

$$SLA(\varphi, \lambda, t) = SL_{Steric}(\varphi, \lambda, t) + SL_{mass}(\varphi, \lambda, t)$$
$$SL_{Steric}(\varphi, \lambda, t) = \frac{1}{\rho_0} \int_{-h}^{0} [\rho(\varphi, \lambda, t, S, T, P) - \overline{\rho}(\varphi, \lambda, \overline{S}, \overline{T}, \overline{P})] dz$$
$$SL_{mass}(\varphi, \lambda, t) = \frac{a_e \rho_e}{3\rho_w} \sum_{n=0}^{96} \sum_{m=0}^{n} \frac{(2n+1)}{(1+k_n)} W_n P_{nm}(\sin \varphi) \times$$
$$\left\{ [\Delta C_{nm}(t) + \Delta C_{nm}^{GAD}(t) - \Delta C_{nm}^{GAA}(t)] \cos(m\lambda) + [\Delta S_{nm}(t) + \Delta S_{nm}^{GAD}(t) - \Delta S_{nm}^{GAA}(t)] \sin(m\lambda) \right\}$$

(3)

Here, $\rho_0$ is the mean seawater density (1028 kg/m³), $h$ is the maximum depth, $\rho$ is the density as a function of geographical latitude ($\varphi$) and longitude ($\lambda$), $t$ is the observational epoch, $S$ is the salinity, $T$ is the temperature, and $P$ is the pressure (for more details, refer to [11]). $\overline{\rho}$, $\overline{S}$, $\overline{T}$, and $\overline{P}$ represent the mean seawater density, mean salinity, mean temperature, and mean pressure, respectively. Additionally, $a_e$ represents the Earth's radius, $\rho_e$ is the mean density of the Earth (5517 kg/m³), $\rho_w$ is the density of freshwater (1000 kg/m³), $\Delta C_{nm}$ and $\Delta S_{nm}$ are the dimensionless Stokes coefficients, $\Delta C_{nm}^{GAD}$ and $\Delta S_{nm}^{GAD}$ are the satellite-derived oceanic pressure coefficients, $\Delta C_{nm}^{GAA}$ and $\Delta S_{nm}^{GAA}$ are the satellite-derived geopotential coefficients for nontidal atmospheric effects, $P_{nm}$ is the associated Legendre polynomial of degree n and order m, $k_n$ is the degree (n) of the Love number, and $W_n$ is the Gaussian smoothing filter with a radius of 300 km to mitigate the correlation of north–south stripes and short-wavelength noise in the Stokes coefficients [39].

In the next sections, the four approaches used to combine DT derived from two different methods using hydrography and altimetry data are explained. The focus is on the algorithms, statistical techniques, and mathematical models utilized in the assimilation process. Furthermore, any essential preprocessing steps needed to effectively integrate the datasets are presented.

### 2.3. Assimilation Using Variance Component Estimation (VCE)

Given the availability of two different types of DT obtained from the aforementioned observations (Equations (1) and (2)), it is advantageous to statistically combine these two representations. One approach involves treating the calculation of DT as a least squares problem, where the DT is determined by assigning weights to the observations and employing the least squares variance component estimation method. This method proves effective in estimating the variance of observation [19]. By enhancing the estimation of the observation covariance matrix ($Ql$), the least squares variance component estimation method aims to estimate the solution of the least squares parameter equation. The observation equations, which combine Equations (1) and (2), can be expressed as follows:

$$\underbrace{\begin{bmatrix} SSH_1 - N_1 \\ SLA_1 + MDT_1 \\ SSH_2 - N_2 \\ SLA_2 + MDT_2 \\ . \\ . \\ SSH_n - N_n \\ SLA_n + MDT_n \end{bmatrix}}_{L} = \underbrace{\begin{bmatrix} 1 & 0 & \ldots & 0 \\ 1 & 0 & \ldots & 0 \\ 0 & 1 & \ldots & 0 \\ 0 & 1 & \ldots & 0 \\ . & . & . & . \\ . & . & . & . \\ 0 & 0 & \ldots & 1 \\ 0 & 0 & \ldots & 1 \end{bmatrix}}_{A} \underbrace{\begin{bmatrix} DT_1 \\ DT_2 \\ . \\ . \\ DT_u \end{bmatrix}}_{X}$$

(4)

In the given context, *L* represents the observation vector, *A* is the design matrix, and *x* denotes the vector of unknown parameters (grid points of *DT*). Notably, salinity and

temperature data are accessible as grid points at the regional level using hydrographic observations, Argos observations, and the WOA model. At the same time, *SSH* observations are limited to the satellite track passing through the study area, meaning the number of *SSH* observations may not align with *SLA* observations. To generate the observations suitable for least squares estimation and *DT* calculation, we generate *SSH* observations at other points using the Kriging algorithm [25] and then incorporate them into the observation equations to construct the grid points of *DT*. For the estimation of least squares variance components, $Ql$ is defined as the observation covariance matrix as follows [19,40]:

$$Q_l = \sum_{k=1}^{2} \sigma_k^2 Q_k \tag{5}$$

where $Q_k$ represents the covariance matrices that are determined based on the variances and covariances of the observations. The coefficients $\sigma_k$ (k = 1,2) are the unknown covariance components that need to be determined using the least squares method [40]. The unknown coefficients $\sigma_k$ are called variance components and are calculated accompanied by the *DT* as unknown parameters in the least squares process. The variance components reflect the effects associated with the dynamic topography obtained from altimetry (with variance $\sigma_{Alt}^2$) and the DT determined by the combination of steric and non-steric sea level anomalies (by variance $\sigma_{Steric+Non-Steric}^2$) in the observation vector. Therefore, assuming we can write [19]:

$$
\begin{aligned}
Q_l &= \sigma_1^2 Q_{Alt} + \sigma_2^2 Q_{Steric+non-Steric} \\
Q_{Alt} &= \begin{bmatrix} \sigma_{SSH1}^2 + \sigma_{N1}^2 & 0 & \cdots & 0 \\ 0 & \sigma_{SSH2}^2 + \sigma_{N2}^2 & \cdots & 0 \\ . & . & . & . \\ 0 & 0 & . & \sigma_{SSHn}^2 + \sigma_{Nn}^2 \end{bmatrix}_{n \times n} \\
Q_{Steric+non-Steric} &= \begin{bmatrix} \sigma_{SLA1}^2 + \sigma_{MDT1}^2 & 0 & \cdots & 0 \\ 0 & \sigma_{SLA2}^2 + \sigma_{MDT2}^2 & \cdots & 0 \\ . & . & . & . \\ 0 & 0 & 0 & \sigma_{SLAn}^2 + \sigma_{MDTn}^2 \end{bmatrix}_{n \times n}
\end{aligned}
\tag{6}
$$

To create a covariance matrix for $Q_{Alt}$, time series of SSH are generated at each satellite altimetry observation point considering the altimetry repeat cycle, and the mean SSH is subtracted from them. Afterward, $\sigma_{SSH}^2$ is designated as the standard deviation of the residual signal at each individual point and added to the value of $\sigma_2^N$ obtained from the Balmino error toolbox [28]. For $Q_{Steric+non-Steric}$, time series of *SLA* are shaped and monthly means are taken from them (e.g., mean of January from January), and the desired standard deviation of the residual signal represents $\sigma_{SLA}^2$. The variance of *MDT* can be computed based on the variances of *MSS* and *N*. Additionally, it is presupposed that the cross-covariance between errors is negligible. Nonetheless, this assumption is not implausible, given that the covariance components are relatively small [25].

To apply this method, one should start with an initial guess for the variance components ($\sigma_{Alt}^2$ and $\sigma_{Steric+Non-Steric}^2$). Through an iterative process, the variance components and, subsequently, the covariance matrix of observations ($Ql$) are computed until the differences between the initial approximation and the estimated variance components [40] tend toward zero. Afterward, DT and its corresponding covariance matrix are determined as described below [40]:

$$
\begin{aligned}
\hat{x} &= (A^T Q_l^{-1} A)^{-1} A^T Q_l^{-1} l \\
C_{\hat{x}} &= (A^T Q_l^{-1} A)^{-1}
\end{aligned}
\tag{7}
$$

### 2.4. Assimilation Using Bayesian Theory Method

Bayesian theory is a mathematical framework used to combine two types of data. In this theory, it combines prior information with new data to generate the final result. Bayesian theory uses Bayes' rule to adjust the probability of an event based on previous

evidence. In the context of data assimilation, Bayesian theory can update posterior probabilities by taking into account prior probabilities. Each type of data is weighted based on prior probabilities. The posterior probability obtained from Bayes' rule can be used to estimate the combined data. The combined data can be calculated by weighted summation or by using a probabilistic model that includes posterior probabilities. Combining data using Bayesian methods provides a logical framework for combining and decision-making based on all available data. This approach is useful when dealing with incomplete or conflicting information as it allows for consistent combination and decision-making based on all available data. Generally, the combination of two types of data by Bayesian theory is as follows [41].

$$
\begin{aligned}
\text{Pr1} &= L1 \times P1 \\
\text{Pr2} &= L2 \times P2 \\
DT_{final} &= \text{Pr1} \times DT_{ALT} + \text{Pr2} \times DT_{Steric+non-steric}
\end{aligned}
\tag{8}
$$

Here, $P1$ represents the prior or initial event for the first data ($DT$ obtained from satellite altimetry), $L1$ is the likelihood of the first data, $P2$ represents the prior or initial event for the second data ($DT$ obtained from steric and non-steric of SLA), $L2$ is the likelihood of the second data, $Pr1$ represents the posterior event for the first data, $Pr2$ represents the posterior event for the second data, $DT_{ALT}$ is the $DT$ obtained from satellite altimetry, $DT_{Steric+non-steric}$ is the $DT$ obtained from steric and non-steric $SLA$, and $DT_{final}$ is the combined dynamic topography.

### 2.5. Assimilation Using Kalman Filter (KF)

The Kalman filter theory is another method that can be used to combine two types of data. The Kalman filter theory operates based on two main steps: prediction and correction. The equations associated with the combination of two types of data using the Kalman filter in an iterative process are as follows [42].

$$
\begin{aligned}
&\text{Prediction step} \\
&\hat{x} = Sx \\
&J = SJS' + D \\
&\text{Update step} \\
&K = JH' \left( HJH' + (Q_{ALT} + Q_{Steric+non-steric}) \right)^{-1} \\
&J = (I - KH) J \\
&L = [DT_{ALT}; DT_{Steric+non-steric}] \\
&\hat{x} = \hat{x} + K (L - H\hat{x}) \\
&DT_{final} = \hat{x}
\end{aligned}
\tag{9}
$$

where $x$ represents the state vector, $S$ is the state matrix, $D$ is the covariance matrix of noise, $I$ is the identity matrix, $J$ is the covariance matrix of the state, $H$ is the observation operator, $K$ is the Kalman gain matrix, and $L$ is the observation vector, $x$ which represents the unknown estimate.

### 2.6. Assimilation Using 3DVAR (3D Variational) Method

The 3DVAR method serves not only as a data assimilation technique but also possesses the capability to integrate diverse data from multiple sources. This method establishes an objective function, taking into account existing errors, and minimizes it to achieve the optimal solution. The resulting solution, obtained by minimizing the objective function, is expressed as follows [19,43].

$$
\begin{aligned}
B &= \left( I + H' (Q_{ALT} + Q_{Steric+non-steric}) H \right)^{-1} \\
\hat{x} &= B \left( B \, DT'_{Alt} + H' Q_{ALT} DT'_{Alt} + H' Q_{Steric+non-steric} DT'_{Steric+non-steric} \right)
\end{aligned}
\tag{10}
$$

In the above equation, *B* is the covariance matrix of the model or background that is derived by utilizing the covariance matrices estimated through the VCE method, and *H* is the observation operator [44]. *H* is a mathematical function that maps the model's state variables to the space of observations. Therefore, here the matrix *H* is derived from matrix A (Equation (4)).

*2.7. Estimation of Total Surface Current*

The total surface current is obtained via summing Ekman surface current and geostrophic current [45]. The Ekman component is sourced from the National Oceanic and Atmospheric Administration (NOAA). Geostrophic currents are derived by computing the zonal (*u*) and meridional (*v*) velocity components through the horizontal gradient of the DT, as described by [38,45]. These components can be calculated using the following equations:

$$
\begin{aligned}
u &= -\frac{g}{f}\frac{\partial \zeta}{\partial y} \\
v &= \frac{g}{f}\frac{\partial \zeta}{\partial x}
\end{aligned}
\tag{11}
$$

where *g* represents the acceleration of gravity (9.832 m/s²), $\zeta$ is *DT*, $f = 2\Omega sin\varphi$ represents the Coriolis force, $\varphi$ and $\Omega$ are geographical latitude and the Earth's rotation speed, respectively [38].

Finally, the total surface current can be estimated as follows [19,38]:

$$
\begin{aligned}
u_{Total} &= u_{Geostrophic} + u_{Ekman}, \quad v_{Total} = v_{Geostrophic} + v_{Ekman} \\
W_{Total} &= \sqrt{u_{Total}^2 + v_{Total}^2}
\end{aligned}
\tag{12}
$$

**3. Results**

In this section, the emphasis is on presenting and analyzing the results. The assimilation technique is evaluated by comparing it with independent measurements. The influence of assimilation hydrography observations and altimetry data into the estimation of DT accuracy is emphasized, and any potential uncertainties or limitations are addressed.

As previously stated, the objective of this section is to calculate DT by utilizing satellite altimetry data along with the steric and non-steric terms of SLA. To accomplish this, we follow a series of steps to derive the final DT:

1.  DT is determined by employing satellite altimetry and integrating the steric and non-steric components of sea surface anomalies.
2.  Two different types of estimated DT are assimilated using the aforementioned approaches.
3.  The final DT is validated by comparing it with local current meter data.

The initial step in determining DT involves the estimation of the mean dynamic topography (MDT) and geoid height in the study area, employing Equations (1) and (2). To evaluate the accuracy of the geoid height calculations, Figure 2 presents the errors associated with gravity models employed in this study. Notably, the XGM2019e model exhibits the lowest error, particularly at higher resolutions, for deriving the geoid height from gravity models.

Within the spatial scale of 80-100 km (equivalent to spherical harmonic degree *n* = 200–250), the geoid derived from this model demonstrates a nominal error of less than 1 cm (Figure 2). Overall, both the XGM2019e gravity model and the EIGEN_6C4 gravity model yield superior outcomes when compared with other gravity models. Figure 3 visually depicts the geoid height obtained from the XGM2019e gravity model in the Persian Gulf and the Sea of Oman. In this particular region, the geoid height ranges from 82 to −1 m, with the minimum value situated in the southeastern part of the Sea of Oman near India and the highest value is situated in the northwestern boundaries of the Persian Gulf.

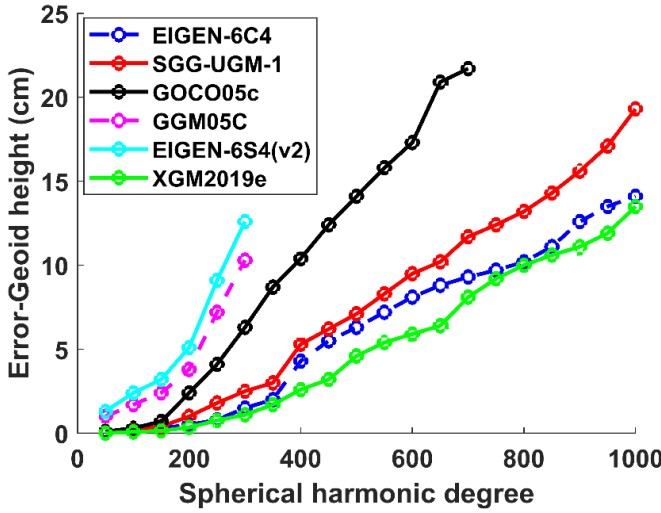

**Figure 2.** The error of geoid height models.

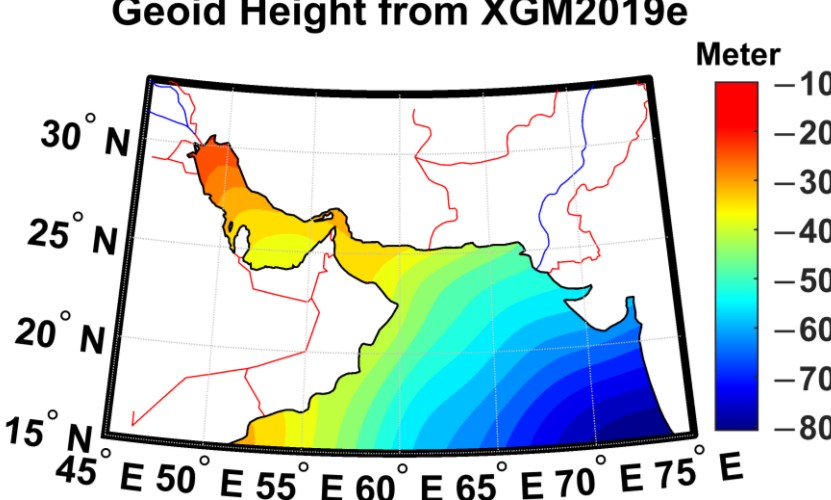

**Figure 3.** Geoid height of XGM2019e gravity model.

The MDT for the study area has been estimated as gridded points with a geographical resolution of 0.25 degrees in longitude and latitude. This estimation is based on the difference in the geoid height between the XGM2019e model and the mean sea surface (MSS-IR01) model. Figure 4a provides a visual representation of the spatial pattern of the unfiltered MDT in the study area. It is important to note that the MDT contains noise due to the disparity in resolutions between the geoid data and the MSS model. In order to reduce this noise, a spatial filter is implemented on the MDT. Prior studies have utilized Gaussian filters with various radii. For instance, [14] employed a Gaussian filter with a 140 km radius, [46] used a 125 km radius Gaussian filter, and [38] applied a 100 km radius filter. Additionally, [47] estimated the size of the MDT filter derived from data obtained from the gravity field and steady-state ocean circulation explorer (GOCE) satellite, suggesting a range of 100 to 125 km for global applications. Taking into account these studies, we have chosen to employ a Gaussian filter with a 100 km radius, considering the geoid height error (less than 1 cm in 100 km resolution) in the XGM2019e model. The filtered MDT is depicted in Figure 4b. The MDT exhibits a range of values spanning from 42 to 78 cm, with higher values predominantly observed toward the southeast and east. Meanwhile, there are lower values observed in the vicinity of the northwestern and western regions.

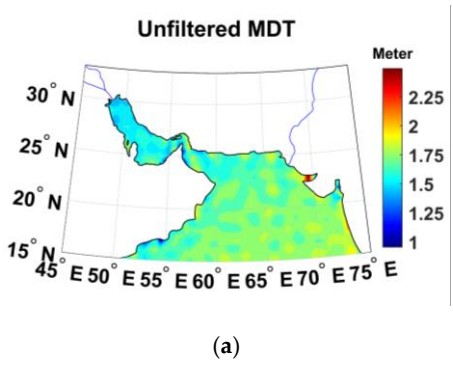
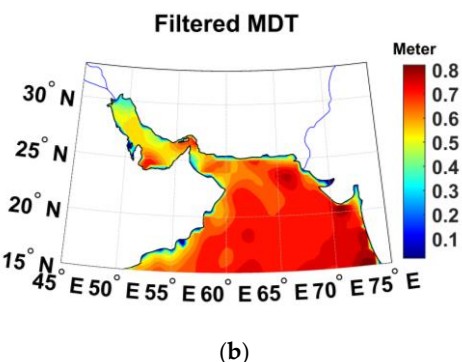

**Figure 4.** (**a**) Unfiltered MDT; (**b**) filtered MDT.

The calculated MDT, derived from the respective errors of MSS-IR01 and XGM2019e, exhibits an error range of mainly 1.2 to 3.4 cm. To assess the accuracy of our MDT model, we compare it with the MDT-CNES-CLS2018 model. The results indicate that both models depict a very similar pattern of MDT within the study area, with variations occurring along the coastlines. Figure 5 provides a visual representation of the MDT-CNES-CLS2018 model and the discrepancy comparing the estimated MDT and the MDT-CNES-CLS2018 model. It is evident that the dissimilarities between the two models arise from the utilization of different gravity models and MSS data. It is essential to highlight that, in the context of this study, we employed the local MSS-IR01 model to calculate the MDT.

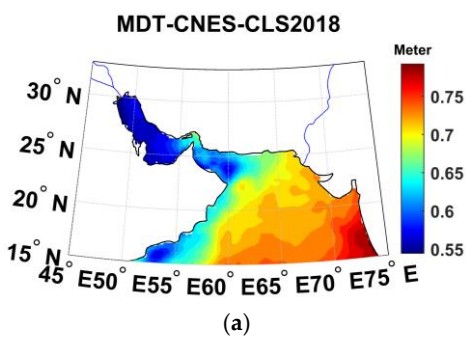
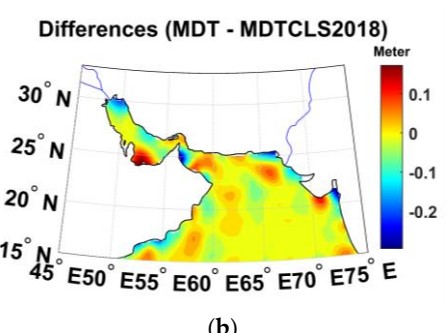

**Figure 5.** (**a**) MDT of CNES; (**b**) differences between our MDT and MDT of CNES.

The calculation of the MDT and geoid within the study area has been completed, serving as crucial components in determining DT through Equations (1) and (2). It is important to note that satellite altimetry observations in coastal regions exhibit lower accuracy compared with offshore areas, even by applying retracking correction. Hence, to achieve a suitable balance between observations, hydrographic observations (steric term) and GRACE satellite data (non-steric term) are employed in determining DT. This entails assigning a relatively higher weight to hydrographic and GRACE satellite observations in coastal areas compared with satellite altimetry observations, while satellite altimetry observations carry greater weight in offshore regions during the final determination of the DT.

Figure 6a,b presents the standard deviation of satellite altimetry observations before and after retracking correction, respectively. As depicted, the standard deviation increases in coastal regions and decreases in offshore regions. Upon comparing Figure 6a,b, it is evident that applying the retracking correction results in a decrease in the standard deviation of satellite altimetry observations in coastal regions. Figure 7 displays the standard deviation of steric and non-steric sea surface height anomaly observations. These standard deviations are employed to construct the initial covariance matrix ($Q_{Alt}$ and $Q_{Steric+non-Steric}$) in the determination of the final DT, utilizing the methods described in

later sections. Moreover, Figure 8 illustrates the SLA for a specific location in the Sea of Oman, using satellite altimetry observations as well as steric and non-steric observations. Notably, the steric and non-steric SLAs exhibit good agreement with satellite altimetry observations. As an example, Figure 9 displays the DT obtained from satellite altimetry data and the combination of hydrographic (steric) and GRACE (non-steric) data for January 2002. The DT data points are generated at a monthly temporal resolution and a spatial resolution of 0.25 degrees. Our current objective is to determine the final DT by utilizing these two types of observations.

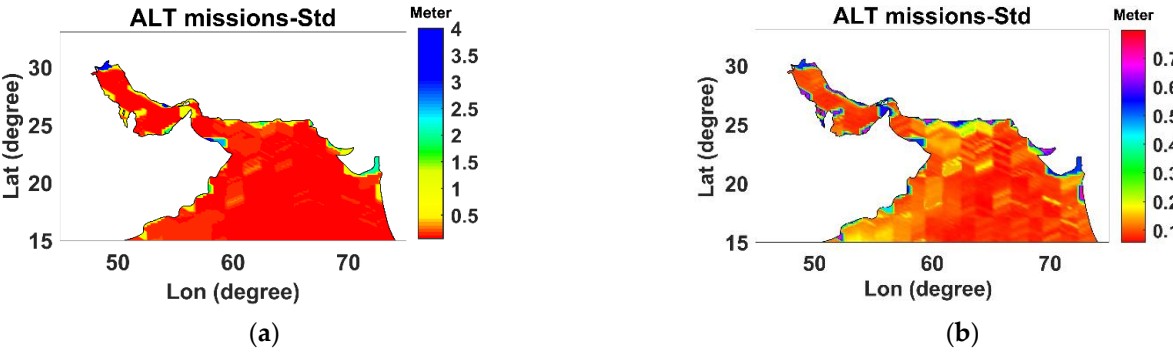

(**a**)                                                                                  (**b**)

**Figure 6.** (**a**) Standard deviation of altimetry missions before retracking correction; (**b**) standard deviation of altimetry missions after retracking correction.

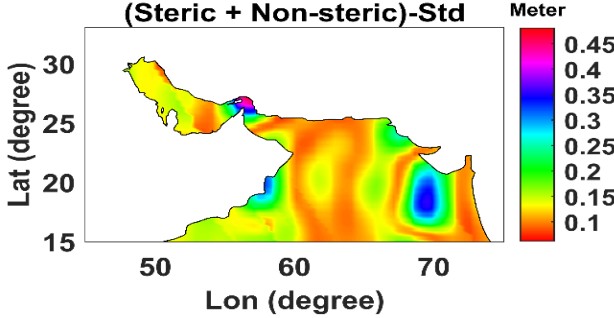

**Figure 7.** The standard deviation of steric + non-steric observations.

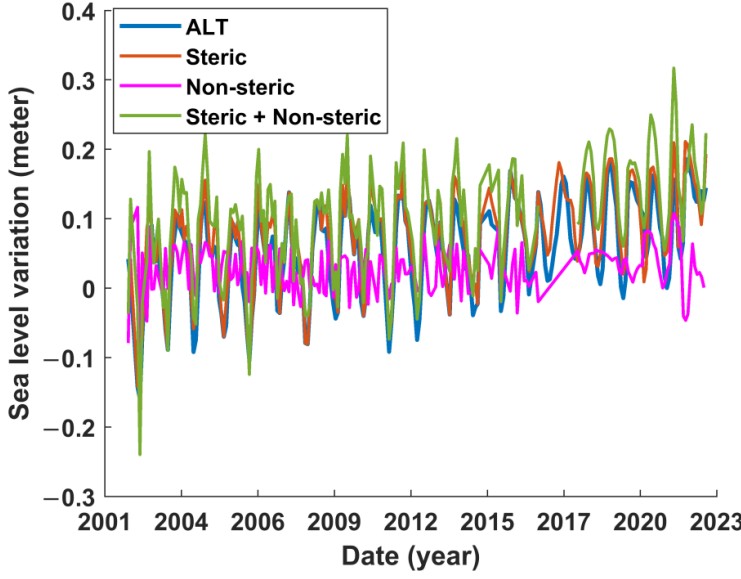

**Figure 8.** SLA of altimetry, steric and non-steric terms of SLA, steric + non-steric (SLA).

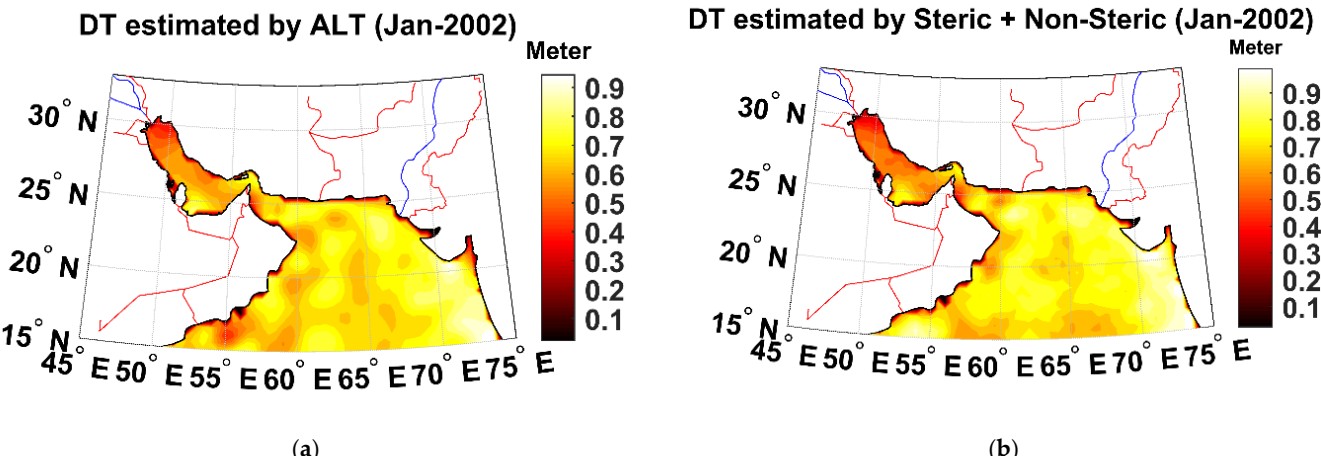

**Figure 9.** (**a**) DT of altimetry; (**b**) DT of steric + non-steric.

For the subsequent step, the observation equations can be expressed in grid points for assimilation, employing methods discussed in the previous section, including VCE, Bayesian theory, Kalman filter, and 3DVAR. Figure 10 provides an example of the DT derived from satellite altimetry observations, as well as steric and non-steric sea surface height anomaly observations. It also showcases the final DT obtained through the application of the aforementioned methods. Most of the methods successfully achieved a satisfactory combination of the two types of data. However, it is challenging to determine the precise optimal method in this particular case.

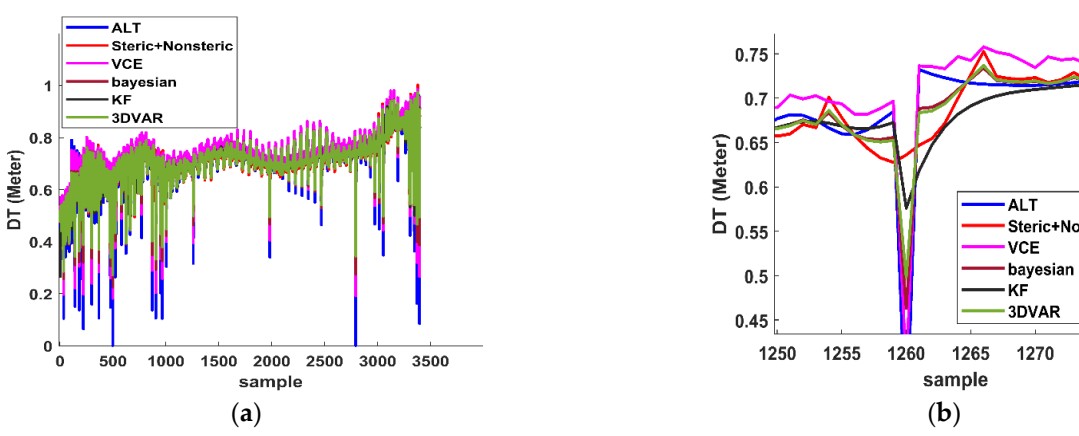

**Figure 10.** The DT resulting from the assimilation of two types of observations using various methods: (**a**) represents the entire sample of the DT while (**b**) shows a specific portion of it.

To validate and determine the suitable method for integrating the two types of observations in the determination of the final DT, local current meter observations (as described in Section 2.1, Table 3) are utilized. The validation process involves computing the total surface currents using the final DT obtained from the previously mentioned methods and comparing them with the local current meter observations. The method that yields the lowest root-mean-square error (RMSE) is chosen as the preferred method. Table 4 provides the RMSE values (in meters per second) between the estimated total surface currents obtained using different DT methods and the current meter observations for the absolute velocity of the oceanic currents. As an illustration, Figure 11 exhibits the surface geostrophic currents derived from the final DT utilizing the aforementioned methods, along with the surface currents at the Konarak station for the surface currents' east–west and north–south components.

**Table 4.** The RMSE values (cm/s) between the estimated total surface currents and the current meter observations for the absolute velocity.

| Station Name | VCE | Bayesian | Kalman Filter | 3DVAR |
|---|---|---|---|---|
| Khoran | 12.15 | 12.25 | 17.30 | 23.65 |
| Konarak | 12.01 | 11.14 | 13.22 | 16.15 |
| Chabahar | 18.23 | 16.33 | 12.41 | 30.41 |
| Bushehr | 13.33 | 17.53 | 14.57 | 41.26 |
| Taheri | 17.21 | 23.34 | 20.43 | 38.55 |
| Nayband | 12.41 | 27.35 | 16.46 | 31.62 |
| Nakhl-Taghi | 10.42 | 12.23 | 10.34 | 26.75 |
| Kangan | 11.44 | 15.32 | 15.30 | 40.54 |
| Jask | 18.32 | 11.12 | 20.43 | 22.19 |
| Larak | 11.47 | 22.52 | 16.55 | 50.53 |
| Googsar | 16.43 | 13.51 | 20.51 | 44.79 |
| Rajaei | 14.20 | 10.36 | 11.52 | 37.31 |

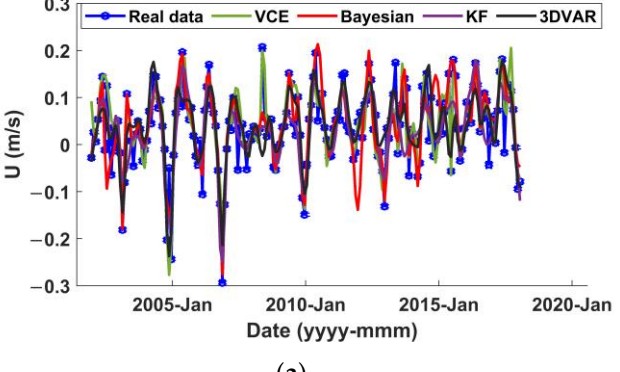

(**a**)

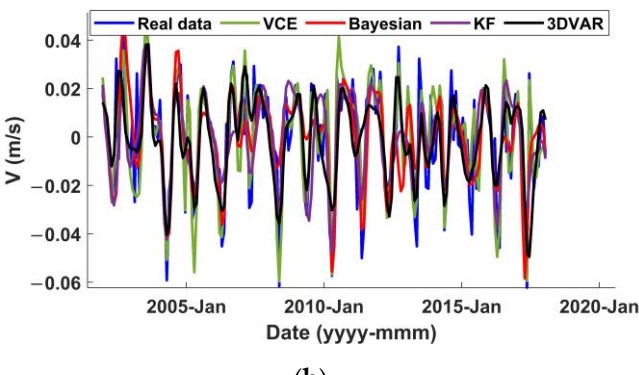

(**b**)

**Figure 11.** Examining the comparison between the estimated total surface currents and the local surface currents specifically at the Konarak station, focusing on: (**a**) the east–west component of the surface currents; (**b**) the north–south component of the surface currents.

Through the comparison of RMSE values calculated between the local current meter observations and the currents estimated using the DT obtained from the four aforementioned methods, it is generally observed that the VCE method exhibits lower RMSE values compared with the other methods at most stations. However, it is worth noting that other methods also performed well at specific stations. As an illustrative example, Figure 12 showcases the final DT obtained utilizing the VCE method for January 2002.

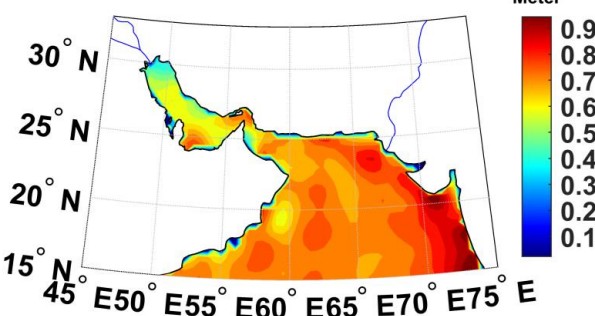

**Figure 12.** Final DT by VCE.

To further examine the impact of integrating steric and non-steric SLA data in determining DT and, subsequently, the total surface currents, we investigate the effect of

combining GRACE and hydrographic data with altimetry observations using the VCE method. The following comparative analysis is conducted:

(i).  Initially, the estimation of total surface currents solely relies on the DT obtained from altimetry observations, without incorporating GRACE and hydrographic data. Then, the estimated currents are compared against the measurements from the current meter.

(ii).  Subsequently, the total surface currents are obtained by utilizing the DT derived from the combined datasets, which include altimetry satellites as well as GRACE and hydrographic data. Furthermore, the estimated currents are compared with the observations from the current meter.

In general, the findings of the study suggest that incorporating both GRACE and hydrographic data into the analysis improves the agreement between the estimated ocean currents and the local observations. Specifically, the VCE approach demonstrates higher accuracy in estimating the currents compared with alternative methods. This indicates that the assimilation of GRACE data and hydrographic measurements enhances the reliability and precision of current estimations. Figure 13 illustrates the differences between the surface current components and the local current meter data in January when considering only altimetry data (without GRACE and hydrographic data) for DT estimation. Conversely, Figure 14 demonstrates the differences comparing the surface current components and the local current meter data in January, taking into account the incorporation of GRACE and hydrographic data for DT estimation. By comparing Figures 13 and 14, it can be observed that the inclusion of the second dataset significantly enhances the determination of the surface currents. In this study, the local current meter stations are located in close proximity to the coast. Acknowledging the well-known limitations of altimetry measurements in coastal regions, the incorporation of GRACE and hydrographic data serves to compensate for these shortcomings, enhancing the accuracy and reliability of the overall analysis.

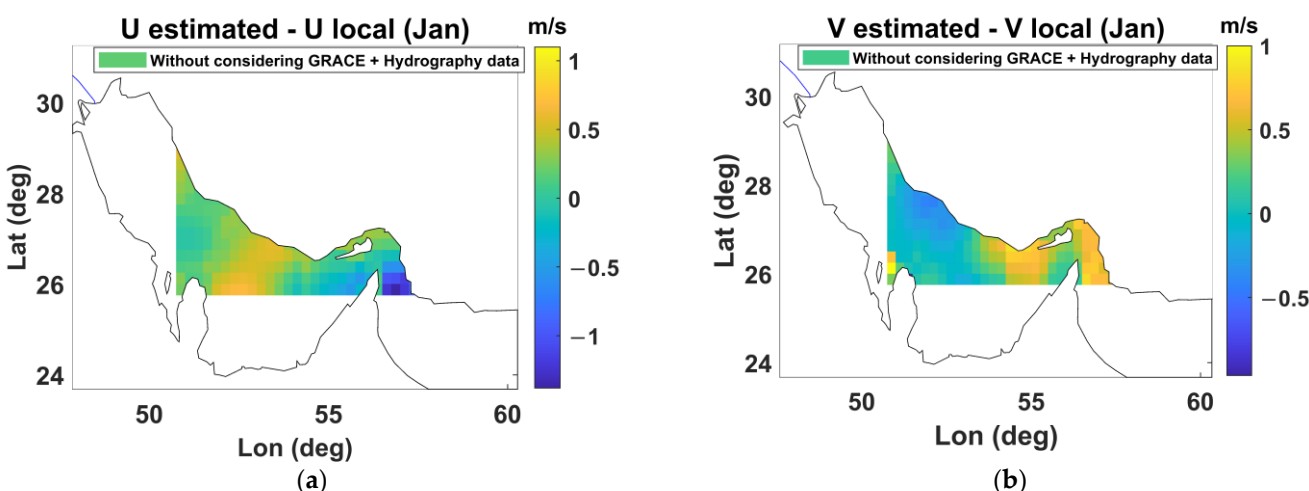

**Figure 13.** The differences between the surface current components (U, V) and the local current meter measurements in January, without considering GRACE and hydrographic data: (**a**) U component (left); (**b**) V component (right).

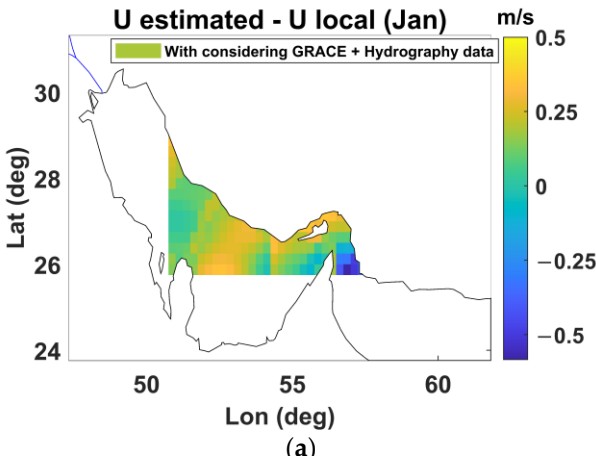
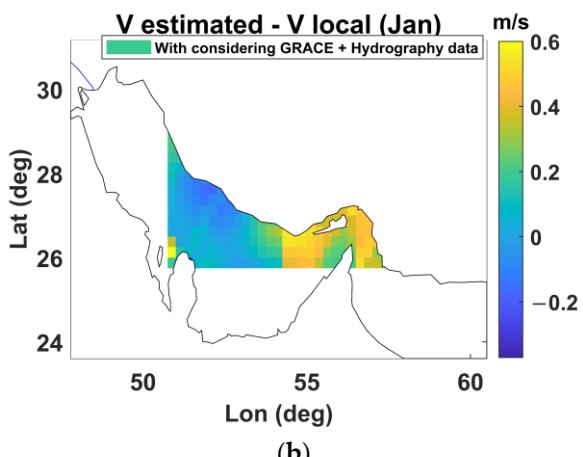

**Figure 14.** The differences between the surface current components (U, V) and the local current meter measurements in January, considering GRACE and hydrographic data: (**a**) U component (left); (**b**) V component (right).

## 4. Discussion

The Persian Gulf and the Sea of Oman hold strategic importance as crucial routes for the transportation of petroleum energy to different parts of the world. Consequently, conducting studies on DT in these regions becomes vital for acquiring a deeper comprehension of the regional climate, environment, maritime transportation, and the dynamics of oil and non-oil pollutants.

The study introduces a novel approach that combines altimetry data with GRACE and hydrographic data, utilizing assimilation methods. This integrated approach aims to estimate DT with higher precision in both coastal and offshore regions, thereby addressing the challenges associated with accurate measurements in coastal areas. The methodology developed in this study exhibits its effectiveness by assigning suitable initial weights to the altimetry and GRACE + hydrographic data, leading to a more dependable estimation of DT. In nearshore, where satellite altimetry observations are known to be less accurate, the GRACE + hydrographic data are given greater weight to compensate for the limitations of altimetry in DT computation. On the other hand, in offshore areas where altimetry performs well, more emphasis is placed on the altimetry data by assigning the data a higher weight. This adaptive weighting strategy ensures that the strengths of each dataset are leveraged appropriately to enhance the overall reliability of DT estimation.

To validate the assimilation methods, the estimated ocean current velocity is compared with and without the inclusion of GRACE (representing the non-steric term of sea level anomalies) and hydrography (representing the steric term of sea level anomalies) data, using in situ current meter data as a reference. The results indicate that incorporating both steric and non-steric terms of sea level anomalies improves the estimation of DT. By combining altimetry and GRACE + hydrographic data through assimilation methods, this study provides a robust framework for obtaining more accurate and reliable estimates of DT, particularly in coastal regions where altimetry observations alone may not suffice.

## 5. Conclusions

The aim of this study is to propose a data-driven approach that determines DT in the Persian Gulf and the Sea of Oman. The proposed method offers the advantage of combining various geodetic data and hydrography observations, such as radar altimetry, GRACE, geoid undulation, salinity, and temperature, using assimilation methods. This approach accurately models the spatial and temporal variations of DT. The DT is then converted into a total surface current by calculating the horizontal gradient (geostrophic current) and incorporating the effects of the Ekman currents to find the appropriate data assimilation method for DT determination.

One of the key conclusions drawn from this study is that the evaluation of different assimilation methods for surface current determination based on DT evaluation reveals that the VCE method outperforms other assimilation approaches, exhibiting lower root-mean-square error (RMSE) by comparing with in situ current meter data. The utilization of the VCE method, which combines altimetry and GRACE + hydrography data, significantly improves the estimated surface currents in coastal and offshore regions.

Another main conclusion is that the inclusion of both the steric and non-steric components of SLA improves the accuracy of DT estimation. Integrating both GRACE and hydrographic data in the analysis enhances the agreement between the estimated ocean currents and the local observations, surpassing the results obtained when only altimetry data are utilized. This suggests that assimilating GRACE data and hydrographic measurements enhances the reliability and precision of current estimations (refer to Figures 13 and 14).

Another key finding is that the MDT and geoid are computed, serving as crucial elements in the determination of DT using Equations (1) and (2). To reduce the noise present in the MDT, a spatial Gaussian filter with a radius of 100 km is applied, yielding favorable results when calculating DT and comparing the estimated MDT with the MDT-CNES-CLS 2018 model. Furthermore, the XGM2019e gravity model demonstrates superior performance in geoid determination compared with other gravity models within the specified spatial scale.

These findings significantly contribute to our understanding of oceanic currents and their dynamics. The proposed method can be applied in future studies and applications that require precise and reliable estimations of surface currents. By improving our ability to monitor and model ocean currents, this research opens up possibilities for advancements in various fields related to oceanography. Future research may focus on refining the assimilation methods used in this study to further improve the accuracy of DT estimation. Exploring the applicability and performance of the proposed method in different geographical regions and under diverse environmental conditions would provide valuable insights. Additionally, ongoing efforts to validate and enhance the accuracy of gravity models and their compatibility with other data sources and models are essential for advancing DT estimation techniques.

**Author Contributions:** conceptualization, M.P. and B.V.; formal analysis, M.P., B.V., D.P. and A.A.; writing—original draft, M.P., B.V., D.P. and A.A.; supervision—review and editing, M.P., B.V., D.P. and A.A. All authors have read and agreed to the published version of the manuscript.

**Funding:** This research is based upon research funded by the Iran National Science Foundation (INSF) under project No. 4014761.

**Data Availability Statement:** The datasets generated and analyzed during the current study are available from the corresponding author upon reasonable request. The data are not publicly available due to the data are part of an ongoing study.

**Conflicts of Interest:** The authors declare no conflicts of interest.

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
