# Peer review of "Integrating Hydrography Observations and Geodetic Data for Enhanced Dynamic Topography Estimation"

_remotesensing, doi:10.3390/rs16030527_

Round 1

Reviewer 1 Report

Comments and Suggestions for Authors

The paper introduces various methods for determining dynamic topography (DT) in the Persian Gulf. The results suggest that their approach successfully determines both DT and current velocities, as evidenced by in-situ data and altimetry measurements. However, some applied methodologies appear to lack thorough validation. Specifically, it may be questionable to derive DT from Sea Level Anomaly (SLA), defined here as steric + ocean mass from GRACE, in addition to Mean Dynamic Topography (MDT). I recommend that the authors meticulously review and organize their methodologies to enhance comprehensibility. 

Some points listed below:

The introduction appears excessively lengthy and lacks sufficient information on the published methods employed for determining dynamic topography (DT) and how the current study aims to advance these approaches.

In lines 29-31, it is recommended to refer to the existing body of published scientific work on the topic.

Line 35 could benefit from an explanation of why data assimilation leads to a more accurate estimation of DT.

Lines 113-121 might be more impactful if moved higher up in the introduction, while ensuring that the remainder of the introduction is streamlined.

Regarding line 147, please add a reference to support the statement.

Concerning Equation 2 (DT = SLA + MDT), clarification is needed on how steric + mass + MDT equals DT. A detailed explanation of the methodology and rationale behind this equation would enhance understanding.

Equation 4 (SLA + MDT = SSH – N) still lacks clarity. It would be beneficial to provide a clear rationale or context for this equation.

For Figures 3 and 4, it is noted that the color bars do not represent the maps in the figures. Adjustments should be made to ensure that the color bars appropriately convey the information in the figures.

Comments on the Quality of English Language

The paper would benefit from a thorough language revision to enhance overall clarity and coherence. Some sections appear unnecessarily lengthy, and there is repetitive elaboration on minor justifications for their research.

Author Response

We would like to thank you for your comments and meticulous check of our manuscript that make the paper more interesting and informative. Below, please find our responses to points raised by the reviewer.

Sincerely Yours

  1. The introduction appears excessively lengthy and lacks sufficient information on the published methods employed for determining dynamic topography (DT) and how the current study aims to advance these approaches.

The introduction has been revised to remove unnecessary content and incorporate additional information regarding previous studies on DT estimation and the connection between the current study and the methods enhancement. Thank you for your comment.

  1. In lines 29-31, it is recommended to refer to the existing body of published scientific work on the topic.

  It was corrected.

  1. Line 35 could benefit from an explanation of why data assimilation leads to a more accurate estimation of DT.

It was corrected.

  1. Lines 113-121 might be more impactful if moved higher up in the introduction, while ensuring that the remainder of the introduction is streamlined.

       It was corrected.

  1. Regarding line 147, please add a reference to support the statement.

It was corrected.

  1. Concerning Equation 2 (DT = SLA + MDT), clarification is needed on how steric + mass + MDT equals DT. A detailed explanation of the methodology and rationale behind this equation would enhance understanding.

More explanation was added to the manuscript.

  1. Equation 4 (SLA + MDT = SSH – N) still lacks clarity. It would be beneficial to provide a clear rationale or context for this equation.

Figure 1 is added to the manuscript to show the connection between oceanic parameters in determining DT.

  1. For Figures 3 and 4, it is noted that the color bars do not represent the maps in the figures. Adjustments should be made to ensure that the color bars appropriately convey the information in the figures.

The figures were corrected.

  1. The paper would benefit from a thorough language revision to enhance overall clarity and coherence. Some sections appear unnecessarily lengthy, and there is repetitive elaboration on minor justifications for their research.

It was corrected.

Reviewer 2 Report

Comments and Suggestions for Authors

The authors of the article unreasonably claim that previous studies by other authors were mainly based on satellite altimetry to determine dynamic topography, which often led to noticeable errors in coastal regions. This is an absolutely incorrect statement. Everyone knows about satellite measurement errors in coastal areas. And no one uses them without correction according to ground-based observations. Normal scientists use data from stationary posts in coastal regions to correct satellite observations. The authors of the article should remove these statements. In addition, it is necessary to completely rewrite the conclusion. In conclusion, it is necessary to give fundamental conclusions and directions for future work that improve the results obtained by the authors of the article.

Author Response

We would like to thank you for your comments and meticulous check of our manuscript that make the paper more interesting and informative. Below, please find our responses to points raised by the reviewer.

Sincerely Yours

The authors of the article unreasonably claim that previous studies by other authors were mainly based on satellite altimetry to determine dynamic topography, which often led to noticeable errors in coastal regions. This is an absolutely incorrect statement. Everyone knows about satellite measurement errors in coastal areas. And no one uses them without correction according to ground-based observations. Normal scientists use data from stationary posts in coastal regions to correct satellite observations. The authors of the article should remove these statements. In addition, it is necessary to completely rewrite the conclusion. In conclusion, it is necessary to give fundamental conclusions and directions for future work that improve the results obtained by the authors of the article.

The incorrect statement had been removed, and the conclusion had also been revised. Thank you for your relevant and impactful comment.

Reviewer 3 Report

Comments and Suggestions for Authors

In this manuscript, the authors combined satellite altimetry and hydrographic data to improve dynamic topography estimation accuracy. Four different assimilation methods have been used. Experiment results through validation using current meter data show that the proposed methods are effective and the variance component estimation achives the best performance. The topic fits the journal. I suggest the authors consider the following problems in revision:

Technical comments:

1.     Add the accuracy values for all the sensor data used.

2.     Lines 266-267, more details about how this is implemented are required. It is not clear.

3.     Line 306, how are B and H obtained?

4.     Eq. (12), the formula for W_total is wrong.

5.     Fig. 3, 5, why are results in (a) and (b) so different?

6.     Line 453, I think the units (m/s) is incorrect for the values in Table 4.

7.     Using current to validate DT results is not straightforward since Ekman current data will also have errors. Why not use buoy data for validating DT directly?

Other comments:

1.     Line 19, correct “which mentioned”.

2.     The variable symbols in the text are different from those in equations, please be consistent.

3.     Line 237, there is no “y” in (4).

4.     Line 315, what is Ï•?

5.     Line 345, correct “and The”.

Comments on the Quality of English Language

Minor corrections are required.

Author Response

We would like to thank you for your comments and meticulous check of our manuscript that make the paper more interesting and informative. Below, please find our responses to points raised by the reviewer.

Sincerely Yours

Technical comments:

  1. Add the accuracy values for all the sensor data used.

The accuracy value of sensors was added to the manuscript.

  1. Lines 266-267, more details about how this is implemented are required. It is not clear.

It was corrected.

In VCE,  ( ) are unknown covariance components which must be determined. The variance components can be determined by the following equation [1]:

In Equation 1, Components of normal matrix and vector  can be obtained with Equation 2 and 3.

In these relations, where:

It should be emphasized that for the above computations, the initial value for  is needed.

To apply this method, one should start with an initial guess for the variance components (  and ). Through an iterative process, the variance components and subsequently the covariance matrix of observations (Ql) are computed until the differences between the initial approximation and the estimated variance components tend toward zero.

**Note that to prevent excessive volume of equations, the aforementioned items were not included in the manuscript.

[1]. Amiri-Simkooei AR, Tiberius CC, Teunissen PJ (2007) Assessment of noise in GPS coordinate time series: methodology and results. J Geophys Res. https:// doi. org/ 10. 1029/ 2006j b0049 13

  1. Line 306, how are B and H obtained?

The matrix B is the model or background covariance matrix. Here, for this matrix, we use the covariance matrix obtained in VCE method for obtaining B ( ). We also examine data assimilation by assigning identity matrix for B and show that using B obtained by VCE is very effective in true identifying the background or model matrix and improves the output of data assimilation.

The observation operator (H) is a mathematical function that maps the model's state variables to the space of observations. It represents the relationship between the model's internal variables and the measurements or observations obtained from real world data sources. The observation operator transforms the model's state variables into the expected observations at specific locations and times. H can be formed based on a physical equation or it can be created based on a mathematical equation such as inverse distance weighted (IDW) interpolation method. The explanation about H is added to the manuscript. The process of obtaining the observation operator depends on the specific application and the relationship between the model variables and the observed quantities.

  1. Eq. (12), the formula for W_total is wrong.

It was corrected.

  1. 3, 5, why are results in (a) and (b) so different?

For Figure 3 (a and b) are different, due to the noise resulting from inconsistencies between the geoid and MSS data resolution. In order to remove the noise, a spatial filter is applied to this MDT. Figure 3a is unfiltered MDT and Figure 3 b is filtered MDT.

Figure 5a and Figure 5b present the standard deviation of satellite altimetry observations before and after retracking correction, respectively. As depicted, the standard deviation increases in coastal regions and decreases in offshore regions. Upon comparing Figure 5a and Figure 5b, it is evident that applying the retracking correction results in a decrease in the standard deviation of satellite altimetry observations in coastal regions.

  1. Line 453, I think the units (m/s) is incorrect for the values in Table 4.

 It was corrected.

  1. Using current to validate DT results is not straightforward since Ekman current data will also have errors. Why not use buoy data for validating DT directly?

Because the study area had limited availability of buoys, which were used to determine the DT, and for validation purposes, independent observations were necessary. Therefore, current meter data as an independent observation was chosen. Furthermore, since the primary objective of determining the DT is to model ocean currents accurately, it would have been more advantageous to utilize current meter data for validation purposes. It is acknowledged that the Ekman current may contain noise, but its impact on the estimation of the total surface current is minimal.

Other comments:

  1. Line 19, correct “which mentioned”.

It was corrected

  1. The variable symbols in the text are different from those in equations, please be consistent.

It was corrected.

  1. Line 237, there is no “y” in (4).

It was corrected.

  1. Line 315, what is Ï•?

It was corrected

  1. Line 345, correct “and The”.

It was corrected

Round 2

Reviewer 1 Report

Comments and Suggestions for Authors

I still observe mismatched colorbars in figures 3 and 12 which raises legitimate concerns about the validity of the presensted results. This discrepancy, where the color schemes of the colorbars differ from those in the actual figures, could indicate either a mistake in preparing the figures or, more concerning, a deliberate manipulation.

A part from the concern above, is the new version of the paper more or less ok. 

Author Response

We would like to express our great appreciation to the reviewer 1 for the time they invested in editing our manuscript and their valuable comments to address their concerns for improving the paper.

Sincerely Yours

I still observe mismatched colorbars in figures 3 and 12 which raises legitimate concerns about the validity of the presensted results. This discrepancy, where the color schemes of the colorbars differ from those in the actual figures, could indicate either a mistake in preparing the figures or, more concerning, a deliberate manipulation.

A part from the concern above, is the new version of the paper more or less ok.

Figures 3 and 12 was corrected.

The error occurred due to the software generating its own default colorbars (Parula) for output, but this issue was resolved.

Reviewer 2 Report

Comments and Suggestions for Authors

The new version of the article looks better and can be published.

Author Response

We would like to express our great appreciation to the reviewer 2 for the time they invested in editing our manuscript and their valuable comments to address their concerns for improving the paper.

Sincerely Yours

The new version of the article looks better and can be published.

Thank you very much

Reviewer 3 Report

Comments and Suggestions for Authors

One minor correct that can be made in the final version:

1. Line 202, "equation 1 and 2" should be "equations 1 and 2".

Author Response

We would like to express our great appreciation to the reviewer 3 for the time they invested in editing our manuscript and their valuable comments to address their concerns for improving the paper.

Sincerely Yours

One minor correct that can be made in the final version:

  1. Line 202, "equation 1 and 2" should be "equations 1 and 2".

It was corrected
